# NOTCH1 promotes the elevation of GM-CSF and IL-6 through the EZH2/STAT3 pathway to facilitate the fibrotic state of the myocardium in DLBCL

**Huimin Zhang** (ORCID)*, **Wen Dong, Lihong Zhang, Bing Ma, Jianying Wang, Yang Shen, Dan Zhao, Wanyi Yin, Yuexian Li, Qingchi Liu**

Hematology Department, The First Hospital of Hebei Medical University, Shijiazhuang, China

* 57703313@hebmu.edu.cn

**Data Availability Statement:** All relevant data are within the manuscript and its Supporting Information files.

## Abstract

### Objective

Investigate the role of the Neurogenic locus notch homolog protein 1 (NOTCH1) signaling pathway in Diffuse large B-cell lymphoma (DLBCL)-related heart pathogenesis.

### Methods

Utilize R (version 4.2.1) to retrieve DLBCL and myocardial infarction datasets from the GEO database, normalize data with limma, perform differential analysis and GO analysis with GOplot, and visualize findings with ggplot2. Various assays were conducted including stable cell line construction, myocardial infarction modeling, imaging, Western Blot, ELISA, staining, and functional assays.

### Results

Significant gene expression and pathway disparities were found between DLBCL and myocardial infarction samples. NOTCH1, The molecules named Recosomal-binding protein 70 (RBP-J), zeste 2 polycomb repressive complex 2 subunit (EZH2), trimethylated histone H3 at lysine 27 (H3K27me3), Signal Transducer And Activator Of Transcription 3 (STAT3) and Jumonji domain containing-3 (JMJD3) matters a lot in DLBCL. NOTCH1 inhibition decreased DLBCL cell proliferation and activity, reduced inflammatory factors, and improved myocardial fibrosis and infarction severity. NOTCH1 inhibits Granulocyte-macrophage colony-stimulating factor (GM-CSF) and Interleukin-6 (IL-6) expressions depending on STAT3 and EZH2. Co-culturing with DLBCL cells increased fibroblast proliferation, invasion, and fibrosis.

### Conclusion

NOTCH1 signaling influences DLBCL development and myocardial infarction severity through the EZH2/STAT3 pathway, leading to increased heart fibrosis.

**Funding:** The author(s) received no specific funding for this work.

**Competing interests:** The authors have declared that no competing interests exist.

## 1. Introduction

In patients with DLBCL, the expression rates of NOTCH1 and Hairy and enhancer of split 1 (Hes1) are significantly elevated compared to the control group, with NOTCH1 expression closely linked to B symptoms, Ann Arbor staging, lymphocyte count, and lactate dehydrogenase levels [1]. NOTCH1 mutations are found in B-cell tumors; and include chronic lymphocytic leukemia/lymphoma, mantle cell lymphoma and diffuse large B-cell lymphoma (DLBCL) [2]. Sidenib also, in a dose- and time- dependent manner, induces cell growth inhibition on DLBCL cell lines down- regulation of NOTCH 1 Gene and Nuclear factor of activated T cells c 1 (NFATC1) in DLBCL cells. and reduces the concentration of Interleukin-10 (IL-10) in the supernatant [3]. Blocking the Notch signaling pathway in macrophages can enhance hepatocyte senescence and promote the reversal of liver fibrosis by upregulating enhancer of EZH2 expression. Inhibition of EZH2 can counteract these effects caused by the blockade of the Notch signaling pathway. RBP-J, as the primary transcription factor of the Notch signaling pathway, interacts with EZH2, inhibiting EZH2's transcriptional activity via Hes1, thereby regulating hepatocyte senescence and liver fibrosis reversal [4]. EZH2 as is known to be associated with methylation chiefly because it functions as the catalytic unit of polycomb repressive complex 2 (PRC2) that is involved in the process of trimethylation of lysine 27 on histone H3 (H3K27) (H3K27me3) [5]. The Notch1/STAT3/Twist signaling axis is implicated in the progression of human gastric cancer, including head and neck squamous cell carcinoma, breast cancer etc [6–8]. Phosphorylation of STAT3 can enhance JMJD3 activity, thereby inhibiting H3K27me3 activity [9,10]. Loss of H3K36me3 leads to the ectopic acquisition of H3K27me3, downregulating Cxadr expression and promoting the overexpression of the PI3K-AKT pathway, Chemokine (C-X-C motif) ligand 1 (CXCL1), and GM-CSF [11]. This article shows that after the CAWS attack, ischemic cardiac fibroblasts release GM-CSF, hence, increasing cardiac inflammation. Mechanistically, GM-CSF stimulates local macrophage chambers to produce inflammatory cytokine and chemokine, and in therapeutics, GM-CSF strongly reduces heart disease [12]. GM-CSF is necessary for the establishment of experimental autoimmune myocarditis (EAM) because it plays the role of the pathogen in the development of the disease. [13].

In summary, there is limited research on cardiac damage in DLBCL. We aim to investigate the exacerbation of cardiac fibrosis by studying the effects of the NOTCH1 signaling pathway through EZH2-induced excessive methylation, resulting in the production of GM-CSF and IL-6, while establishing an animal model of DLBCL with an additional myocardial infarction model.

## 2. Methods

### 2.1 Bioinformatics analysis

Used R (4.2.1) version, R packages: GEOquery [2.64.2], limma [3.52.2], ggplot2 [3.3.6], ComplexHeatmap [2.13.1]. Because the GEO database covered data in multiple fields such as tumors and non-tumors, and the data quality was high and the data types were rich, the GEO database was chosen as the basis for biochemical studies. Using GEOquery package obtained the dataset GSE23501 and dataset GSE48060 from the GEO database, and re-standardized the data through the normalizeBetweenArrays function of the limma package, removing the probes where one probe corresponded to multiple molecules; When interrogating probes associated with the same molecule, he or she retained the probe that had the largest signal value for that species. From the first part of this process, took the intersection of the differentially expressed genes of GSE23501 and GSE48060. and visualized it with a Venn diagram. Used R (4.2.1) version, R packages: clusterProfiler [4.4.4], GOplot [1.0.2], ggplot2 [3.3.6], ID

conversion package: org.Hs.eg.db. Species: Human (Homo sapiens). After ID conversion of the input molecule list, performed enrichment analysis with the clusterProfiler package, and determined the zscore code of every enter of the enrichment with the help for the GOplot package using the provided molecule values. Subsequently, excluded the gene clusters and pathways with biological characteristics depending on the DEGs and then the enrichment study results were visualized with the help of the ggplot2. Processed the data using log2(value + 1), used Spearman statistics, and performed correlation analysis on NOTCH1, RBPJ, EZH2, STAT3 in the data. Visualized the co-expression scatter plot of the analysis results using the ggplot package.

## 2.2 Establishment of stable cell lines

The mRNA sequences for the CDS regions of NTOCH1 and EZH2 were retrieved from NCBI. The human NTOCH1 and EZH2 genes were utilized as models to construct silencing plasmids using the pCDH vector.

For NTOCH1, the oligo sequence was `CCGGCCGGGACATCACGGATCATATCTCGAGATAT GATCCGTGATGTCCCGGTTTTTG`.

For EZH2, the oligo sequence was `CCGGCGGCTCCTCTAACCATGTTTACTCGAGTAA ACATGGTTAGAGGAGCCGTTTTTG`.

A triple-plasmid packaging system comprising pCDH-EV/pCDH-NOTCH1-eGFP-Fluc transfer plasmids, pCDH-sh-EZH2-eGFP-Fluc transfer plasmids, pSPAX2 packaging plasmids, and pMD2.G envelope plasmids was employed.

Initially, 293T cells were seeded in a 6 cm culture dish and the culture medium was refreshed when cell density reached 70% to 80%. Subsequently, 2.5 μg of pCDH-EV/pCDH-EZH2 plasmid, 1.5 μg of pSPAX2, and 1.5 μg of pMD2.G were combined in Opti-MEM to form solution A. In another vessel, 30 μl of DNA transfection reagent PolyJet was mixed with Opti-MEM to create solution B. Solution B was blended into solution A after brief incubation, and the mixture was carefully applied onto the 293T cell culture. The culture medium was replaced after 12 hours and the supernatant was harvested after 48 hours. It was centrifuged at 2000 rpm for 10 minutes, the cell precipitate was removed, and the supernatant was filtered through a 0.45 μm membrane to get the virus containing supernatant fluid. For cell selection, cells were plated in a 6-well plate and treated with varying concentrations of puromycin to establish stable strains. SU-DHL-2 cells were plated in a 6-well plate, infected with virus-containing supernatant, and selected with puromycin for stable cell lines. The mice were positioned in the supine position, and the surgical area on the left chest was shaved followed by cleaning and sterilization. A longitudinal incision of the skin about 1. 5 cm was done in the skin about 1–2 mm from the left edge of the sternum, and suture was used to close the incision using vertical external mattress suture. Superficially, the chest wall muscles were scanned one after the other and the chest cavity was accessed by the 3 rd or 4 th intercostal space, and the intercostal space was propped open with hemostatic forceps. The heart was gently squeezed with the left hand in conjunction with the heartbeat to make the heart pop out of the foramen ovale. The anterior descending branch of the coronary artery was ligated with an 8–0 suture with thread at 1~2mm below the lower edge of the left auricle and 0.5mm next to the cone of the pulmonary artery with appropriate tightness and control of the depth of the needle and the width of the needle, and the ischemia was proven to be successful when the outer surface of the anterior wall of the left ventricle was seen to be pale white. The chest cavity was closed and postoperative recovery was monitored. Silencing plasmids for NOTCH1 and EZH2 were generated, viral particles for cell transduction were produced, stable cell lines were selected, and an MI model was established in mice for further research.

## 2.3 Vivo imaging of small animals

Tumor cell survival and growth in NOD/SCID mice were tracked by inoculating cells labeled with the eGFP-Fluc reporter gene into the tail vein. This approach enabled monitoring of tumor formation, growth, and metastasis. The Fluc signal imaging was started from the second day after tumor cell injection and was performed at 4-day intervals. The in vivo imaging instrument was prepared and initialized. The CCD camera was ensured to be operational once the temperature reached -90˚C. Approximately 150μl of luciferin substrate was aspirated using a sterile 1ml syringe and injected into the mouse's abdominal cavity. The gas anesthesia system was activated, adequate oxygen flow was verified, and sufficient anesthetic dosage was confirmed. 3% isoflurane was used as an anesthetic gas. The luciferin-injected mouse was placed into the gas anesthesia chamber, anesthesia was initiated for about 6 minutes, and the mouse's condition was monitored. The anesthetized mouse was transferred to the sample chamber of the in vivo imaging device, its position was adjusted, the anesthesia gas input was activated, and the chamber door was closed. The camera's scan time was set to 1–30 seconds, the field of view was adjusted, and images were captured. Data were analyzed and processed using living imaging software. At the end of the experiment, the mice were euthanized using CO2. During the euthanasia process, the cages of the mice were ensured to be clean and comfortable, provided with sufficient food and water, and maintained at the appropriate temperature and humidity to reduce the anxiety of the mice. Euthanasia of mice was confirmed to be completed when their heartbeats ceased for 15 minutes. Animal studies used in this particular research were reviewed and granted by the Animal Ethics Committee of The First Hospital of Hebei Medical University. Ethics number: HBYKDX202211180007.

## 2.4 The expression of NOTCH1 and EZH2 was detected by RT-qPCR

The process of the RNA extraction was carried out according to the instructions described by the manufacturer of Trizol reagent and the total RNA was isolated from cells. Next, 200 ng of the isolated total RNA was reverse transcribed into cDNA using a reverse transcription kit. PCR amplification was done with using an RT-qPCR Instrument and reagents with Trans Start Top Green qPCR Super Mix. 2 μl of cDNA was taken for qPCR, and the reaction conditions were as follows: 95˚C for 2 minutes, 95˚C for 15 seconds, 60˚C for 30 seconds, for a total of 40 cycles. Data were quantified using the $2^{-\triangle\triangle Ct}$ method, with GAPDH as an internal reference, to analyze the relative mRNA expression levels of Notch1 and EZH2. NOTCH1: forward primer: 5'-AGGTGCACCCACAGAACTTA-3'; reverse primer: 5'-TCGGACCAATCAGAGATGT T-3'. EZH2: forward primer: 5'-CGCTTTTCTGTAGGCGATGT-3'; reverse primer: 5'-TGGG TGTTGCATGAAAAGAA-3'. GAPDH: forward primer: 5'-ATGACCCCTTCATTGACCTCA-3'; reverse primer: 5'-GAGATGATGACCCTTTTGGCT-3'.

## 2.5 ELISA detection of GM-CSF and IL-6

All reagents were prepared in advance and thoroughly mixed during dilution to avoid bubble formation. To do this, blank wells, standard wells, and test sample wells were prepared. A hundred microliters of sample diluent was mixed with the blank wells while a hundred microliters of the standard or test samples filled the remaining wells. The microplate was placed in a 37˚C incubator and reacted for 120 minutes, ensuring the use of fresh standard solution. The liquid was discarded and the plate was patted dry. Human anti-botin, 100μl of the working solution was added to all the wells and incubated at 37˚C for 60 minutes. The microplate was washed 3 times, immersing each time for 1–2 minutes. Subsequently, 100μl of HRP-labeled avidin working solution was applied and allowed to react at 37˚C for 60 minutes. Detection was performed using an ELISA reader.

## 2.6 Masson staining

Sections were dewaxed in water, and stained with Weigert iron hematoxylin for 5 to 10 minutes, differentiated in acid ethanol for 5 to 15 sec, counterstained with Masson's blue solution for 3 to 5 minutes. The sections were then rinsed in distilled water for 1 minute, stained with 0. 2% Ponceau S solution for 5 to 10 minutes, rinsed with the working solution containing 0. 85% v/v phosphoric acid for 1 minute, washed in the phosphomolybdic acid solution for 1 to 2 minutes, and finally rinsed again with the working solution containing 0. 85% v In aniline blue staining, the sections were stained in staining solution for one to two minutes, and then rapidly dehydrated by treating with 95% ethanol and absolute ethanol three times, each for five to 10 seconds; followed by clearing by xylene for one to two minutes in each of the three changes. Last of all, the sections were closed with the neutral resin.

## 2.7 TCC Staining

The heart was removed, cleaned, and blood clots were removed. It was rinsed with cold saline and fixed in 10% neutral formalin. The heart was washed, dried, and frozen at -20˚C for 15 minutes to harden. The heart was sliced from apex to base into 1mm thick slices. The slices were placed in 5ml 37˚C 1% TTC phosphate buffer solution and incubated for 15 minutes in a water bath. TTC staining showed the infarct area as white, the infarct border zone as brick-red, and the normal area as red.

## 2.8 Immunofluorescence

Mouse myocardial tissue was fixed in 10% neutral formalin, dehydrated, cleared, and embedded to obtain 4μm thick sections. The sections were flattened and adhered to adhesive slides, and baked to secure. The sections were dewaxed, dehydrated, antigens retrieved, washed with TBS, blocked, and then incubated with primary antibodies at 4˚C overnight. Sections were washed and incubated with secondary antibodies labeled with fluorescein for 1 hour at room temperature. They were mounted with DAPI-containing medium, observed, and images captured under a fluorescence microscope.

## 2.9 Western blot

Protein extraction was initiated, treating samples with RIPA lysis buffer and protease inhibitors, then protein content was quantified using the BCA protein concentration determination kit. Subsequently, Protein separation was done by SDS-PAGE as described earlier with the protein being transferred to a PVDF membrane. Post-transfer, the membrane was blocked with 5% skim milk or 5% BSA solution to prevent nonspecific binding. Then, the membrane was incubated with primary antibodies (RBPJ, p-STAT3, EZH2, H3K27me3, GM-CSF, IL-6, a-SMA, Collagen-I, Fibronectin) overnight at 4˚C, washed, and incubated with secondary antibodies for 2 hours at room temperature. Finally, substrate chemiluminescence reaction was performed using ECL reagents and the protein bands were imaged using an exposure machine for densitometric analysis.

## 2.10 Co-culture of cardiac fibroblasts and SU-DHL-2 cells

Human cardiac fibroblasts and SU-DHL-2 cells were obtained from Cytiva. The cells were grown in complete DMEM medium (PM150210) which was later on supplemented with 10% fetal bovine serum (164210–50) and 1% penicillin-streptomycin (PB180120) and incubated at 37˚C in a humidified incubator. The cardiac fibroblasts were put in the lower compartment of

the Transwell insert while the SU-DHL-2 was in the upper compartment. Co-culture was carried out using an indirect co-culture system for 48 hours.

## 2.11 Clonogenic assay

Cardiac fibroblasts were harvested in complete culture medium, seeded in six-well plates at 400 cells per well. They were cultivated until colonies had more than 50 cells. The medium was refreshed every 3 days, and cell status was monitored. The cells were then washed with PBS, fixed with 4% paraformaldehyde, stained with 0.1% crystal violet, washed, air dried and representative images were taken.

## 2.12 CCK8 assay

100 μl of cell suspension per well was seeded in a 96-well plate and pre-incubated at 37˚C with 5% CO2. Then, 10μl of CCK solution was added to each well, avoiding bubble formation, and the plate was returned to the incubator for 1 to 4 hours. The absorbance was measured at 450nm wavelength using a microplate reader.

## 2.13 Transwell assay

Transwell inserts were coated with Matrigel dilution solution, and hydrated at 4˚C. Serum-free culture medium with BSA was prepared. Cells were digested, starved, resuspended in medium, and seeded into inserts. FBS or chemotactic factor was added to the lower chamber of the plate. After 12–48 hours, inserts were removed, fixed with formalin, stained with crystal violet, and photographed for cell counting and analysis.

## 2.14 Statistical analysis

All data was analyzed using GraphPad Prism 9. 0 and expressed in mean ± standard deviation. Independent sample t-test was used to compare the means of two variables in two groups while analysis of variance was used to compare the means of the variable in three or more groups. *P<0.05 was used as an indicator of statistical significance means. Each of the experiments was performed three times in triplicate.

# 3.Results

## 3.1 Bioinformatics analysis

The relevant datasets GSE23501 for DLBCL were screened through the GEO database, and 69 DLBCL-related samples were obtained. The samples were grouped according to the status in the dataset and divided into 7 reference groups and 62 observation groups. The relevant datasets GSE48060 for myocardial infarction were again screened through the GEO database, and 52 myocardial infarction-related samples were obtained. The samples were grouped according to the status in the dataset and divided into 21 reference groups and 31 observation groups. The clustering situation between DLBCL and myocardial infarction samples was observed through the PCA plot (Fig 1A). The thresholds for the DLBCL dataset GSE23501 were set as |log2 Fold Change| > 1 and p -value < 0.05; the thresholds for the myocardial infarction dataset GSE48060 were set as |log2 Fold Change| > 0.5 and p -value < 0.05, and the limma package was used to perform differential analysis between the DLBCL and myocardial infarction sample groups. The differential analysis results were visualized using a volcano plot (Fig 1B), where each point represents a gene, the blue points are significantly down-regulated genes, and the red points are significantly upregulated genes; the black points indicate unaltered expression genes. The intersection genes among the differentially expressed genes of

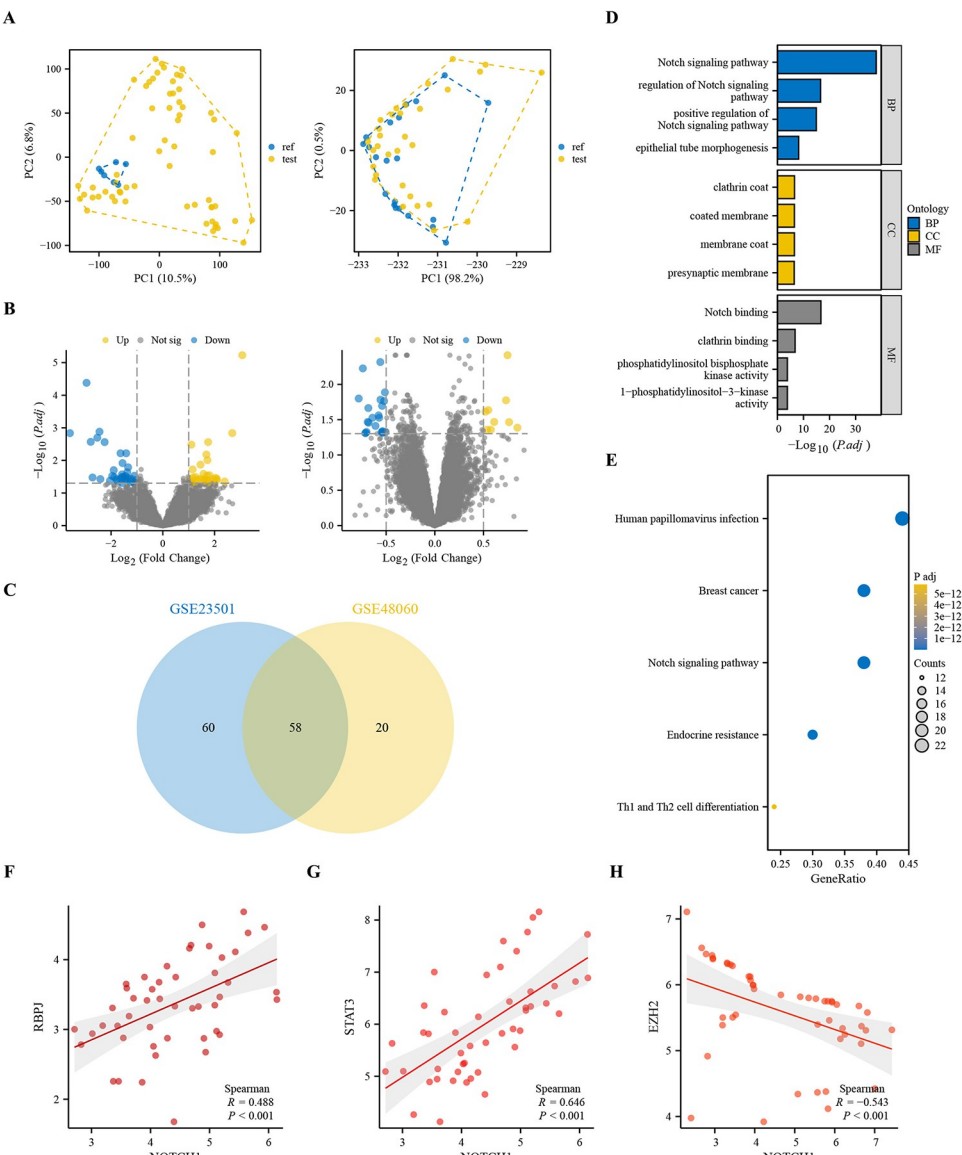

**Fig 1. Analysis of DLBCL and myocardial infarction samples.** (A) PCA plot showing the clustering between DLBCL and myocardial infarction samples. (B) Volcano plot illustrating the results of the differential analysis between DLBCL and myocardial infarction sample groups, with blue points representing significantly downregulated genes, red points indicating significantly upregulated genes, and black points denoting genes with unchanged expression. (C) Heatmap displaying the significantly expressed molecules in DLBCL samples, showcasing the top 50 differentially expressed genes with red representing significantly upregulated genes and blue representing significantly downregulated genes. (F) Co-expression heatmap highlighting the values of NOTCH1 and showing the changing trends of RBPJ, STAT3, and EZH2 relative to NOTCH1. (G-J) Co-expression scatter plots demonstrating the relationship between gene expressions, including correlations between NOTCH1 and RBPJ, RBPJ and EZH2, EZH2, and H3K27, and STAT3 and JMD3 with gene expressions indicated in red and fitted lines in red.

GSE23501 and GSE48060 (Fig 1C). The differentially expressed genes corresponding to the GO analysis were analyzed to integrate the GO terms and create a biological process network of differentially expressed genes. GO divides the results of functional annotation into three categories: biological process (BP), cellular component (CC), and molecular function (MF). After visualizing the results, the GO histogram (Fig 1D) and the KEGG bubble plot (Fig 1E) were

obtained. GO and KEGG analyses were performed through ClusterProfiler, and it was found that DEGs were significantly enriched in multiple biological pathways. GO analysis showed that DEGs were significantly enriched in biological processes such as the Notch signaling pathway, regulation of Notch signaling pathway, and positive regulation of Notch signaling pathway. KEGG pathway analysis revealed that DEGs were significantly enriched in the Notch signaling pathway, TNF signaling pathway, and B cell receptor signaling pathway. The gene co-expression scatter plot uses red to represent the expression level of the gene and red to represent the fitting line. The NOTCH1 and RBPJ co-expression relationship scatter plot (Fig 1F) was drawn, and the correlation coefficient between NOTCH1 and RBPJ was 0.488; the NOTCH1 and STAT3 co-expression relationship scatter plot (Fig 1G) was drawn, and the correlation coefficient between NOTCH1 and STAT3 was 0.646; the NOTCH1 and EZH2 co-expression relationship scatter plot (Fig 1H) was drawn, and the correlation coefficient between NOTCH1 and EZH2 was -0.543.

## 3.2 NOTCH1 modulates the oncogenicity of DLBCL and the severity of myocardial infarction by suppressing EZH2 gene

By transducing lentiviruses encoding Notch1 and EZH2, SU-DHL-2 cells derived from DLBCL were infected. The cells were divided into three groups and injected into NOD/SCID mice through the tail vein to establish a DLBCL mouse model followed by myocardial infarction induction. The three groups were: the LV-KD-NC group, LV-KD-NOTCH1 group and LV-KD-NOTCH1-KD-EZH2 group. In order to detect the lentiviral infection efficiency, we used RT-qPCR experiments for validation, and the results showed that the relative mRNA expression of NOTCH1 was significantly lower and that of EZH2 was significantly higher in the LV-KD-NOTCH1 group relative to the LV-KD-NC group; and that there was no significant difference in the relative mRNA expression of NOTCH1 in the LV-KD-NOTCH1-KD-EZH2 group relative to the LV-KD-NOTCH1 group, there was no significant difference in the relative mRNA expression of NOTCH1 and a significant decrease in the relative mRNA expression of EZH2 in the LV-KD-NOTCH1-KD-EZH2 group. This result suggests that NOTCH1 knockdown was successful and the knockdown efficiency of NOTCH1 was 58.64% (Fig 2A).

After transduction with firefly luciferase-labeled SU-DHL-2 stable cells, lymphoma tissues exhibited significant fluorescent signals, indicating high expression of firefly luciferase in lymphoma cells. The intensity and distribution of fluorescence correspond to the location and size of lymphoma, reflecting the activity and proliferation of lymphoma cells. Compared to the LV-KD-NC group, the fluorescence intensity and counts significantly decreased in the LV-KD-NOTCH1 group; compared to the LV-KD-NOTCH1 group, the fluorescence intensity and counts increased in the LV-KD-NOTCH1-KD-EZH2 group (Fig 2B).

ELISA was performed to measure the expression levels of GM-CSF and IL-6 in serum. Compared to the LV-KD-NC group, the expression levels of GM-CSF and IL-6 inflammatory factors decreased significantly in the LV-KD-NOTCH1 group; compared to the LV-KD-NOTCH1 group, the expression levels of GM-CSF and IL-6 inflammatory factors increased significantly in the LV-KD-NOTCH1-KD-EZH2 group (Fig 2C).

## 3.3 NOTCH1 modulates the severity of myocardial infarction in DLBCL model

Masson's trichrome staining revealed that compared to the LV-KD-NC group, the degree of myocardial fibrosis was significantly reduced in the LV-KD-NOTCH1 group, with fewer blue-stained areas indicating mild myocardial lesions. Compared to the LV-KD-NOTCH1 group,

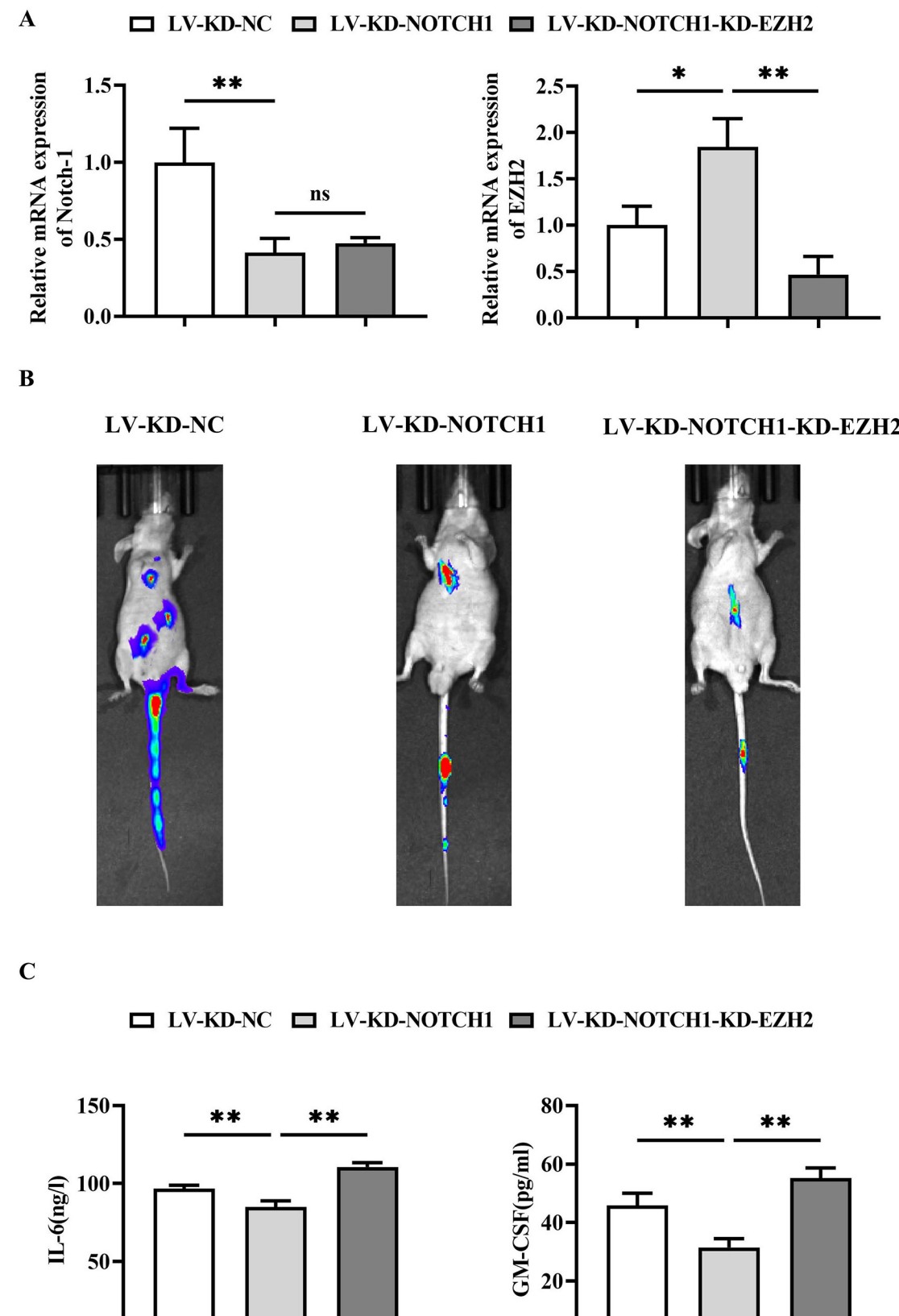

**Fig 2. NOTCH1 modulates the oncogenicity of DLBCL and the severity of myocardial infarction by suppressing EZH2 gene.**
A: Statistical plots of relative mRNA expression of NOTCH1 and EZH2 in LV-KD-NC group, LV-KD-NOTCH1 group and LV-KD-NOTCH1-KD-EZH2 group by RT-qPCR; B: In vivo imaging of lymphoma tissues in LV-KD-NC group, LV-KD-NOTCH1 group and LV-KD-NOTCH1-KD-EZH2 group, depicting the expression of fluorescence signals; C: ELISA analysis of the expression levels of GM-CSF and IL-6 in the serum of LV-KD-NC group, LV-KD-NOTCH1 group, and LV-KD-NOTCH1-KD-EZH2 group. Data are expressed as mean±SD. N = 6; LV-KD-NC group vs LV-KD-NOTCH1 group, **P<0.01; LV-KD-NOTCH1 vs LV-KD-NOTCH1-KD-EZH2, **P<0.01.

the degree of myocardial fibrosis was significantly increased in the LV-KD-NOTCH1-KD-EZH2 group, with more blue-stained areas indicating severe myocardial lesions (Fig 3A).

TTC staining showed that compared to the LV-KD-NC group, there were fewer white infarct areas in the LV-KD-NOTCH1 group. Collagen fiber staining was mild, and the distribution was more sparse. Compared to the LV-KD-NOTCH1 group, there were more white infarct areas in the LV-KD-NOTCH1-KD-EZH2 group. Collagen fiber staining intensified, and the distribution became denser (Fig 3B).

Subsequent tissue immunofluorescence staining revealed the marker Collagen-I in stained myocardial fibroblasts. Compared to the LV-KD-NC group, the fluorescence intensity of Collagen-I was significantly reduced in the LV-KD-NOTCH1 group. Compared to the LV-KD-NOTCH1 group, the fluorescence intensity of Collagen-I increased significantly in the LV-KD-NOTCH1-KD-EZH2 group (Fig 3C).

## 3.4 NOTCH1 inhibits the production of GM-CSF and IL-6 via the STAT3 and EZH2 pathways

Subsequently, cells were divided into seven groups and stimulated with incrementally increasing inhibitors. Seven groups were delineated: LV-KD-NC group, LV-KD-NOTCH1 group, LV-KD-NOTCH1+stattic (STAT3 inhibitor, 10μM) group, LV-KD-NOTCH1-KD-EZH2 +stattic group, NC group, Stattic group and Colivelin group (STAT3 activator, 50μg/mL).

Compared to the LV-KD-NC group, the expression of RBPJ significantly decreased in the LV-KD-NOTCH1 group, LV-KD-NOTCH1+stattic group, and LV-KD-NOTCH1-KD-EZH2 +stattic group;

Compared to the LV-KD-NC group, the expression of p-STAT3 significantly decreased in the LV-KD-NOTCH1 group, LV-KD-NOTCH1+stattic group, and LV-KD-NOTCH1-KD-EZH2+stattic group, with weak expression;

Compared to the LV-KD-NC group, the expression of EZH2 and H3K27me3 significantly increased in the LV-KD-NOTCH1 group and LV-KD-NOTCH1+stattic group. The LV-KD-NOTCH1-KD-EZH2+stattic group exhibited weak expression;

Compared to the LV-KD-NC group, the expression of GM-CSF and IL-6 significantly decreased in the LV-KD-NOTCH1 group and LV-KD-NOTCH1+stattic group, with weaker expression. The LV-KD-NOTCH1-KD-EZH2+stattic group showed a significant increase in the expression of GM-CSF and IL-6, with pronounced elevation (Fig 4).

## 3.5 Enhanced proliferative and invasive properties of myocardial fibroblasts co-cultured with DLBCL

SU-DHL-2 cells derived from DLBCL were co-cultured with myocardial fibroblasts for 48 hours. The co-culture medium and myocardial fibroblast medium were mixed at a ratio of 1:10 to serve as the base culture medium for the colony formation experiment of myocardial fibroblasts. Clonal formation experiments revealed that compared to the LV-KD-NC group, the number of colonies formed significantly decreased in the LV-KD-NOTCH1 group, and further decreased in the LV-KD-NOTCH1+stattic group. Compared to the LV-KD-NC group,

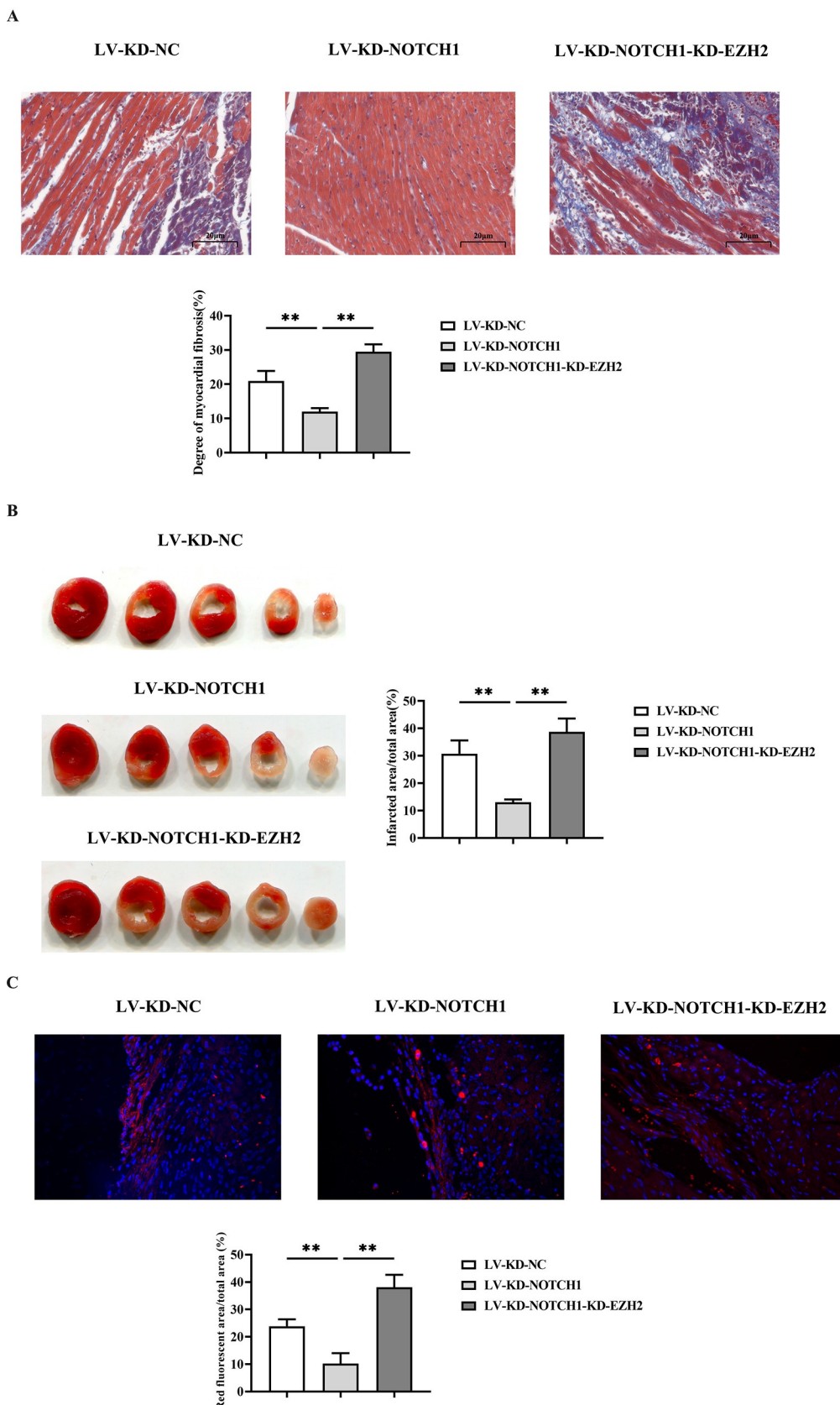

**Fig 3. NOTCH1 modulates the severity of myocardial infarction in DLBCL model.** A: Masson staining was used to detect the degree of fibrosis in myocardial tissue of the LV-KD-NC, LV-KD-NOTCH1 group, and LV-KD-NOTCH1-KD-EZH2 group; B: TCC staining detecting the infarct area in the LV-KD-NC group, LV-KD-NOTCH1 group, and LV-KD-NOTCH1-KD-EZH2 group; C: Immunofluorescence staining of tissue revealing Collagen-I fluorescence intensity in LV-KD-NC group, LV-KD-NOTCH1 group, and LV-KD-NOTCH1-KD-EZH2 group. N = 6; Data are expressed as mean±SD. LV-KD-NC group vs LV-KD-NOTCH1 group, **P<0.01; LV-KD-NOTCH1 vs LV-KD-NOTCH1-KD-EZH2, **P<0.01.

the number of colonies formed significantly increased in the LV-KD-NOTCH1-KD-EZH2+-stattic group. The number of colonies formed by the Colivelin group was significantly higher relative to the NC group, whereas the number of colonies formed by the Stattic group was significantly lower (Fig 5A).

Through Transwell experiments, it was revealed that compared to the LV-KD-NC group, the number of cells traversing the basement membrane significantly diminished in the LV-KD-NOTCH1 group. Furthermore, in contrast to the LV-KD-NOTCH1 group, the LV-KD-NOTCH1+stattic group exhibited an even more pronounced reduction in the number of cells traversing the basement membrane. Conversely, when compared to the LV-KD-NC group, the LV-KD-NOTCH1-KD-EZH2+stattic group demonstrated a notable increase in the number of cells traversing the basement membrane. Relative to the NC group, the number of cells crossing the basement membrane was significantly higher in the colivelin group, whereas the number of cells crossing the basement membrane was significantly lower in the Stattic group (Fig 5B).

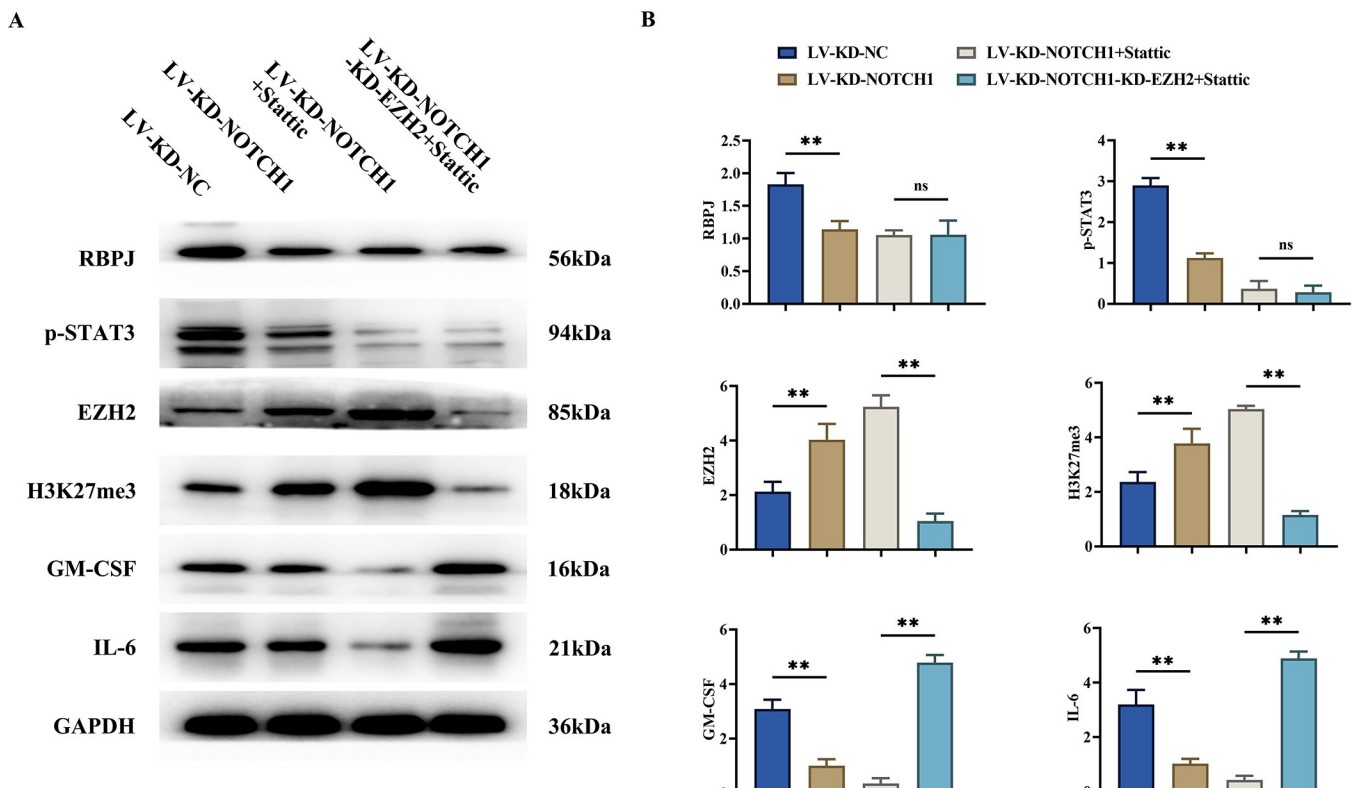

**Fig 4. NOTCH1 inhibits the production of GM-CSF and IL-6 via the STAT3 and EZH2 pathways.** A: Western blot analysis of protein banding plots for RBPJ, p-STAT3, EZH2, H3K27me3, GM-CSF and IL-6; B: Western blot analysis of the relative protein expression of RBPJ, p-STAT3, EZH2, H3K27me3, GM-CSF and IL-6. GAPDH was used as a control protein. Data are expressed as mean±SD. N = 3; LV-KD-NC group vs LV-KD-NOTCH1 group, **P<0.01; LV-KD-NOTCH1+stattic group vs LV-KD-NOTCH1-KD-EZH2+stattic group, **P<0.01, nsP>0.05.

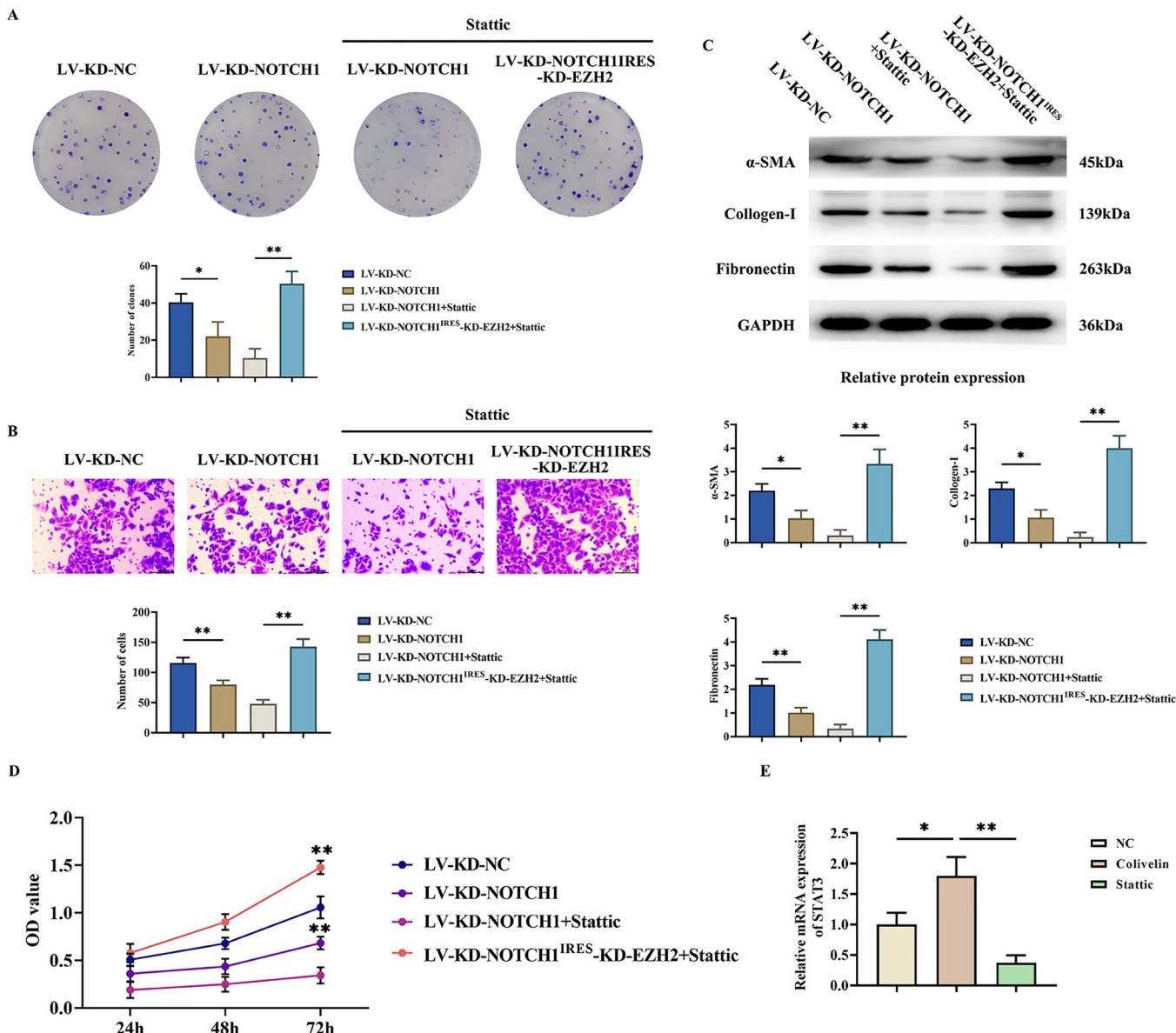

**Fig 5. Enhanced proliferative and invasive properties of myocardial fibroblasts Co-cultured with DLBCL.** A: Clonogenic Assay results graph and clone counts; B: Graph of results of Transwell experiments and statistics of the number of fibroblasts crossing the basement membrane; C: Western blot analyses of the strip charts of a-SMA, Collagen-I and Fibronectin and changes in relative protein expression levels, GAPDH was used as a control protein; D: CCK8 detects the OD of each group at 450 nm. E: Plot of the results of RT-qPCR to detect the relative mRNA expression of STAT3 in NC group, Colivelin group and Stattic group. Data are expressed as mean±SD. N = 3; LV-KD-NC group vs LV-KD-NOTCH1 group, **P<0.01, *P<0.05; LV-KD-NOTCH1+stattic group vs LV-KD-NOTCH1-KD-EZH2+stattic group, **P<0.01, [ns]P>0.05; NC group vs Stattic group, **P<0.01; NC group vs Colivelin group, **P<0.01; Stattic group vs Colivelin group, **P<0.01, [ns]P>0.05.

Through Western Blot experiments, it was discerned that in comparison to the LV-KD-NC group, the protein expression levels of a-SMA, Collagen-I, and Fibronectin notably decreased in the LV-KD-NOTCH1 group. Similarly, in comparison to the LV-KD-NOTCH1 group, the LV-KD-NOTCH1+stattic group exhibited significant reductions in the protein expression levels of a-SMA, Collagen-I, and Fibronectin. Conversely, when contrasted with the LV-KD-NC group, the LV-KD-NOTCH1-KD-EZH2+stattic group demonstrated a marked increase in the protein expression levels of a-SMA, Collagen-I, and Fibronectin (Fig 5C).

CCK8 assays indicated that compared to the LV-KD-NC group, the OD value at 450nm significantly decreased in the LV-KD-NOTCH1 group and further decreased in the LV-KD-NOTCH1+STAT3 inhibitor Statistic group. Compared to the LV-KD-NC group, the OD value at 450nm significantly increased in the LV-KD-NOTCH1-KD-EZH2+stattic group. Relative to NC group, Colivelin group showed significantly higher OD at 450 nm while Stattic group showed significantly lower OD at 450 nm (Fig 5D). And the results of RT-qPCR experiments showed that the relative mRNA expression of STAT3 was significantly higher in the Colivelin group and lower in the Stattic group relative to the NC group (Fig 5E).

### 3.6 A schematic diagram depicting the role of Notch1 via the EZH2 and H3K27 pathways in cardiac fibroblast modulation

NOTCH1 can inhibit the expression of EZH2 by restraining RBPJ through the NOTCH1 pathway receptor, which in turn promotes the downregulation of GM-CSF and IL-6 via H3K27 methylation inhibition. Elevated levels of GM-CSF and IL-6 in the serum can foster the proliferation and differentiation of cardiac fibroblasts, thereby facilitating the generation of cardiac fibrosis (Fig 6).

## 4. Discussion

DLBCL combined with myocardial infarction, a rare primary cardiac tumor comprising 1% of cases, presents diagnostic and treatment challenges due to varied cardiac symptoms. Treatment with anthracycline drugs in R-CHOP regimens raises cardiac toxicity concerns, with factors like age, diabetes, and hypertension influencing risk. Risk classification models aid in predicting cardiac toxicity in DLBCL patients. Long-term studies show comparable cardiac toxicity between high-dose and standard regimens. DLBCL patients face cardiovascular risks post-anthracycline chemotherapy, emphasizing the need for monitoring and prevention strategies. Treatment complexities for DLBCL patients with cardiac involvement highlight the need for further research in reducing cardiac toxicity risks [14–17].

In this study, utilizing the GEO database, we identified the DLBCL-related dataset GSE23501, comprising 69 DLBCL-related samples partitioned into 7 reference groups and 62 observation groups. Simultaneously, we selected the myocardial infarction-related dataset GSE48060, encompassing 52 myocardial infarction-related samples distributed among 21 reference groups and 31 observation groups. A principal component analysis (PCA) plot was employed to discern the clustering patterns among DLBCL and myocardial infarction samples. Differential analysis between these groups was executed utilizing the limma package, visually elucidated through a volcano plot, where blue points denote genes significantly downregulated, yellow points signify genes significantly upregulated, and black points represent genes with unaltered expression levels. Additionally, a heatmap was crafted to visualize the markedly expressed molecules in DLBCL samples, showcasing the top 50 differentially expressed genes, where and up arrow represents significantly up-regulated genes and down arrow represents down regulated genes. The above differentially expressed genes were subjected to the GO analysis later on in order to combine GO terms and establish a biological process network. The enriched pathways in Biological Process (BP), Cellular Component (CC), and Molecular Function (MF) of the top 12 differentially expressed genes, as identified in GOKEGG enrichment analysis, encompass processes such as chemical synaptic transmission, cell adhesion, and potassium ion transport. Furthermore, a co-expression heatmap delineated the values of NOTCH1 and visually depicted the fluctuating trends of RBPJ, STAT3, and EZH2 about NOTCH1. Co-expression scatter plots were generated to illustrate the correlation between gene expressions, including a plot depicting the relationship between NOTCH1 and RBPJ,

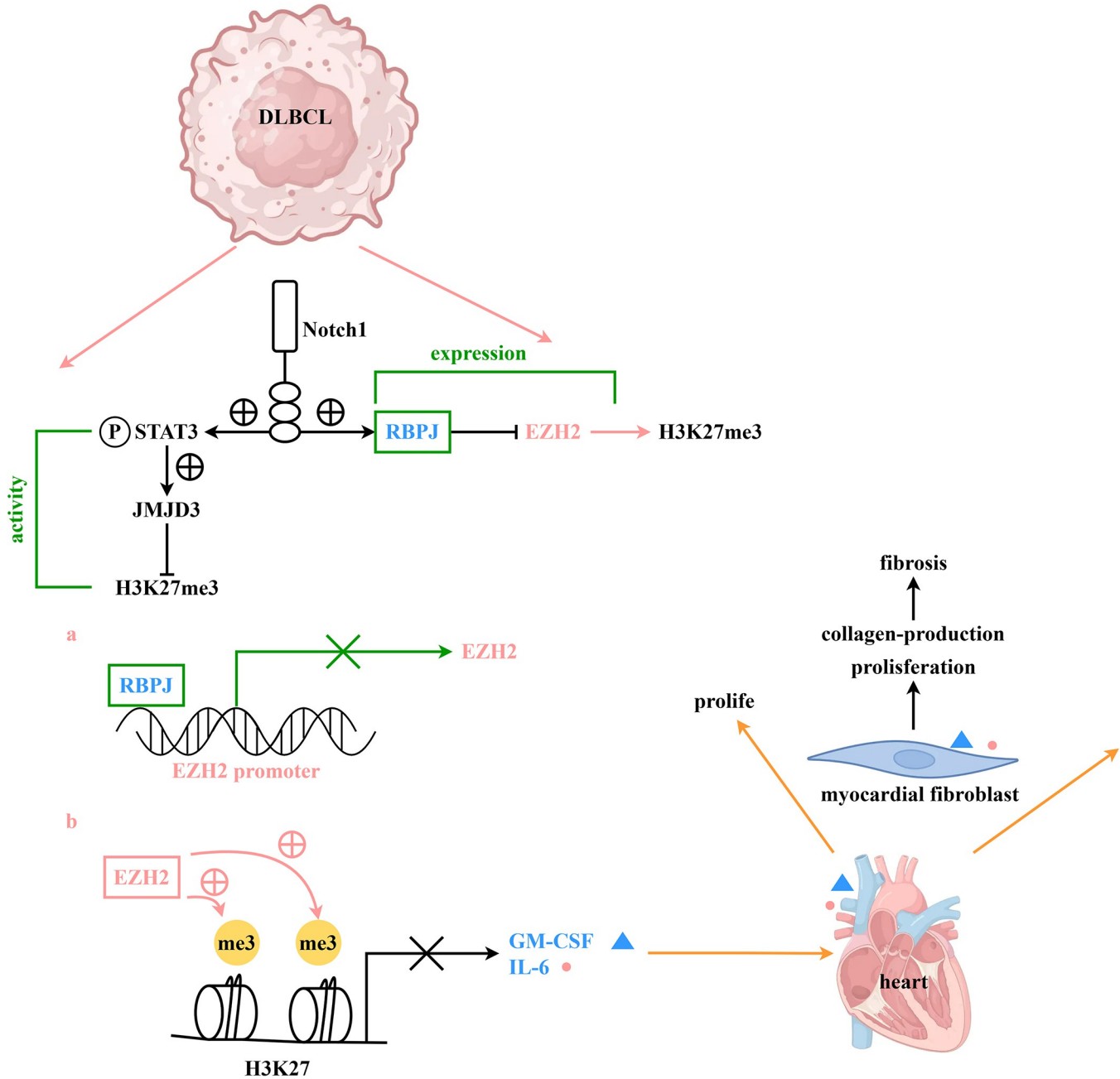

**Fig 6. Schematic representation of Notch1 acting on cardiac fibroblasts via the EZH2 and H3K27 pathway.** The NOTCH1 pathway receptor RBPJ inhibits the expression of EZH2, thereby affecting the expression levels of GM-CSF and IL-6, ultimately influencing the proliferation, differentiation, and fibrosis of cardiac fibroblasts.

NOTCH1 and EZH2, and NOTCH1 and STAT3, with gene expressions indicated in red and fitted lines in red. a DLBCL mouse model was created by lentivirus-mediated knockdown of NOTCH1 and EZH2 genes, followed by myocardial infarction induction. Results showed decreased lymphoma cell activity and proliferation in the NOTCH1 knockdown group compared to the control group. Myocardial cell fibrosis was also reduced in the NOTCH1 knockdown group, suggesting NOTCH1's role in DLBCL tumorigenesis and cardiac injury. Notch1

activation mutations contribute to treatment resistance in DLBCL, especially in Richter syndrome, impacting disease invasiveness and chemotherapy resistance. Aberrant Notch1 signaling in B cells affects T cell function, making Notch1 a potential target for B cell tumor treatment. Notch1 mutations in certain DLBCL subtypes are linked to patient prognosis, with elderly patients having poorer PFS and OS with Notch1 mutations [18–21].

Notch1 mutations in DLBCL, these mutations may be driving factors in the pathogenesis of DLBCL [22]. In this study, NOTCH1 was found to suppress EZH2 gene expression, impacting DLBCL tumorigenicity and myocardial infarction extent. In mouse models, NOTCH1 inhibition decreased lymphoma cell activity and proliferation, as indicated by reduced fluorescence intensity and distribution. NOTCH1 inhibition also lowered GM-CSF and IL-6 levels in serum, while simultaneous EZH2 inhibition increased their expression. NOTCH1 inhibition decreased myocardial fibrosis and infarction area, while EZH2 inhibition increased fibrosis. NOTCH1 suppressed GM-CSF and IL-6 production through STAT3 and EZH2 pathways. NOTCH1 inhibition reduced myocardial fibroblast properties, while EZH2 inhibition enhanced them. Notch1 signaling plays a role in immune evasion in DLBCL, impacting treatment strategies. EZH2 can upregulate CXCR4, suggesting a role in DLBCL stem cell marker expression [23]. Tazemetostat and Belinostat targeting EZH2 and HDAC simultaneously affect the immunogenicity of GC-DLBC [24].Inhibition of EZH2 by SHR2554 and HDAC by Chidamide demonstrated combinational anti-tumor action in DLBCL as they lessen DNA replication procedures inclusive of ORC1 [25,26]. Since the results using ChIP assay showed that STAT3 was bound to the Jmjd3 promoter, Jmjd3 can be considered as direct target of STAT3 [27]. Here in the study, Jmjd3 over expression suppressed glioblastoma stem cell proliferation and neurosphere formation and Jmjd3 knockdown reversed the neurosphere formation deficits induced by STAT3 inhibitors. Shi and colleagues' studies showed that suppression of STAT3 caused histone H3K27 demethylation of the neuron differentiation genes such as Myt1, FGF21, and GDF15 among others. Activation of NFκB by JMJD3 in wound macrophages promoted the inflammatory gene expression through H3K27me3 and elevated GM-CSF and IL-6 levels. There was an up-regulation of GM-CSF in cardiac fibroblasts from veins and atria within hours after CAWS attack which promoted inflammation of the cardiac tissue. End-stage heart failure tissues showed increased GM-CSFR expression in myocardial cells. Plasma GM-CSF and adhesion molecule levels were notably high in patients with AMI and severe left ventricular dysfunction [28,29].

In this study, GM-CSF and IL-6 were found to be crucial in DLBCL progression and cardiac injury. NOTCH1 inhibition significantly reduced their expression, potentially alleviating DLBCL-related cardiac inflammation. Interestingly, co-suppression of EZH2 led to a resurgence of GM-CSF and IL-6, suggesting a collaborative role with NOTCH1 in regulating inflammation. Further experiments with the stattic highlighted changes in RBPJ, p-STAT3, EZH2, and H3K27me3 levels in the NOTCH1 inhibition group. Combined NOTCH1 and STAT3 inhibition notably decreased GM-CSF and IL-6 expression, shedding light on how NOTCH1 regulates inflammatory factors through the STAT3 and EZH2 pathways [30,31].

Flavo is cardioprotective because it lowers cTn-I, NF-κB, APP, and GM-CSF; the latter is through the GM-CSF/NF-κB signaling, which has antioxidant properties, anti-inflammatory, and anti-apoptotic activities [32]. NOTCH1 inhibition was found to decrease proliferation, invasiveness, and fibrosis of cardiac fibroblasts in co-culture, clonogenic, CCK8, and Transwell experiments. This suggests that NOTCH1 may indirectly impact cardiac injury by influencing cardiac fibroblast function. Targeting the NOTCH1 pathway could be beneficial in treating DLBCL and cardiac injury, offering potential therapeutic advantages. Research focusing on NOTCH1's molecular mechanisms in DLBCL should be pursued to improve treatment

strategies and minimize cardiac toxicity. Overall, NOTCH1 plays a complex role in DLBCL and cardiac injury, with its inhibition providing new therapeutic possibilities.

## Supporting information

**S1 Table. Tabular data.** Note: This document contains all the tabular data generated in the paper.
(XLS)

**S2 Table. Raw bands from western blotting.** Note: This file contains the original bands obtained in the western blot experiments generated by the study.
(PDF)

**S1 File.**
(PDF)

## Author Contributions

**Data curation:** Huimin Zhang, Wen Dong, Lihong Zhang, Bing Ma, Jianying Wang, Yang Shen, Dan Zhao, Wanyi Yin, Yuexian Li, Qingchi Liu.

**Formal analysis:** Jianying Wang.

**Writing – original draft:** Huimin Zhang, Wen Dong, Lihong Zhang, Bing Ma, Yang Shen, Dan Zhao, Wanyi Yin, Yuexian Li, Qingchi Liu.

**Writing – review & editing:** Huimin Zhang, Wen Dong, Lihong Zhang, Bing Ma, Yang Shen, Dan Zhao, Wanyi Yin, Yuexian Li, Qingchi Liu.

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
