## [Decision Letter · Decision Letter 0]

24 Jun 2024

PONE-D-24-21738NOTCH1 promotes the elevation of GM-CSF and IL-6 through the EZH2/STAT3 pathway to facilitate the fibrotic state of the myocardium in DLBCLPLOS ONE

Dear Dr. Zhang,

Thank you for submitting your manuscript to PLOS ONE. After careful consideration, we feel that it has merit but does not fully meet PLOS ONE’s publication criteria as it currently stands. Therefore, we invite you to submit a revised version of the manuscript that addresses the points raised during the review process.Please provide necessary information in Methods and Figure Legend.Please have careful proofreading to reduce grammatical errors. Please adequately address the reviewers's concerns.Please submit your revised manuscript by Aug 08 2024 11:59PM. If you will need more time than this to complete your revisions, please reply to this message or contact the journal office at plosone@plos.org. Please include the following items when submitting your revised manuscript:A rebuttal letter that responds to each point raised by the academic editor and reviewer(s). You should upload this letter as a separate file labeled 'Response to Reviewers'.A marked-up copy of your manuscript that highlights changes made to the original version. You should upload this as a separate file labeled 'Revised Manuscript with Track Changes'.An unmarked version of your revised paper without tracked changes. You should upload this as a separate file labeled 'Manuscript'.

We look forward to receiving your revised manuscript.

Kind regards,

Meijing Wang, MD

Academic Editor

PLOS ONE

Journal Requirements:

   "Funding Title: Key Technology Research Program (Hebei Provincial Health Committee)

Research Topic: Application Study of Flow Cytometry Fluorescent Immunodetection of Cytokines in Patients with Malignant Hematologic Diseases

Funding Number: 20221410"

5. In the online submission form, you indicated that "All data generated or analyzed during this study are included in this published article. Additional datasets analyzed during the current study are available from the corresponding author upon reasonable request."

9. PLOS ONE now requires that authors provide the original uncropped and unadjusted images underlying all blot or gel results reported in a submission’s figures or Supporting Information files. This policy and the journal’s other requirements for blot/gel reporting and figure preparation are described in detail at https://journals.plos.org/plosone/s/figures#loc-blot-and-gel-reporting-requirements and https://journals.plos.org/plosone/s/figures#loc-preparing-figures-from-image-files. When you submit your revised manuscript, please ensure that your figures adhere fully to these guidelines and provide the original underlying images for all blot or gel data reported in your submission. See the following link for instructions on providing the original image data: https://journals.plos.org/plosone/s/figures#loc-original-images-for-blots-and-gels.   

Reviewers' comments:

Reviewer's Responses to Questions

**Comments to the Author**

1. Is the manuscript technically sound, and do the data support the conclusions?

Reviewer #1: Partly

Reviewer #2: Partly

2. Has the statistical analysis been performed appropriately and rigorously? 

Reviewer #1: No

Reviewer #2: Yes

3. Have the authors made all data underlying the findings in their manuscript fully available?

Reviewer #1: Yes

Reviewer #2: Yes

4. Is the manuscript presented in an intelligible fashion and written in standard English?

Reviewer #1: No

Reviewer #2: Yes

5. Review Comments to the Author

Reviewer #1: In this manuscript, the authors investigated the role of the NOTCH1 signaling pathway in DLBCL-related heart dysfunction. They found that NOTCH1 inhibition decreased DLBCL cell proliferation, reduced the inflammatory response, and attenuated myocardial fibrotic remodeling and infarction severity following ischemic injury. However, there are several issues that need to be addressed:

1. The manuscript contains numerous writing errors and requires significant improvement in scientific writing.

2. Detailed methods regarding the heart ischemic injury model need to be included.

3. The immunofluorescence staining of tissue does not appear to be of heart tissue in Figure 3c.

4. Several important details are missing, such as the dose of the STAT3 inhibitor and the sample number. Additionally, the efficiency of shRNA is not shown.

Overall, this manuscript is not well written and lacks necessary information. The conclusions are not well supported by the provided data. Therefore, I recommend a major revision to address these issues.

Reviewer #2: In this study, Zhang et al. investigated the role of the NOTCH1 signaling pathway in the pathogenesis of DLBCL-related heart disease. Through data mining and bioinformatic analysis of DLBCL and myocardial infarction GEO databases, they identified the activation of signaling pathways including NOTCH, STAT3, and EZH2, underscoring their pivotal roles in the pathogenesis of DLBCL-related heart disease. However, several issues and concerns need to be addressed:

1. The authors should justify the rationale for comparing the DLBCL and myocardial infarction GEO databases to identify signaling pathways contributing to the pathogenesis of DLBCL-related heart disease.

2. For the bioinformatic analysis, the authors need to describe the criteria for selecting the database and how the samples were categorized into reference and observation groups.

3. The co-expression scatter plots in Figure 1 G-J do not support the claimed correlations between RBPJ and EZH2, EZH2 and H3K27, and STAT3 and JMD3, except NOTCH1 and RBPJ.

4. In Figure 5 A-D, the results for Static treatment alone should be presented to conclusively determine the synergistic effects of NOTCH and STAT3 signaling inhibition on fibroblast growth.

5. The manuscript appears to be prepared hastily, containing numerous grammatical errors. Additionally, the figure legends lack detail and do not always correspond to the figures presented. For instance, in the legend for Figure 1, there is no distinction between panels 1A-a and 1A-b; there is no description for Figures 1D and 1E. Furthermore, Figure 1C contains two panels both labeled as Figure 1C-a. Also, many abbreviations are used without being defined upon first use.

6. PLOS authors have the option to publish the peer review history of their article (what does this mean?). If published, this will include your full peer review and any attached files.

Reviewer #1: No

Reviewer #2: No

---

## [Author Response · Author response to Decision Letter 0]

6 Sep 2024

PONE-D-24-21738

NOTCH1 promotes the elevation of GM-CSF and IL-6 through the EZH2/STAT3 pathway to facilitate the fibrotic state of the myocardium in DLBCL

PLOS ONE

Dear Dr. Zhang,

Thank you for submitting your manuscript to PLOS ONE. After careful consideration, we feel that it has merit but does not fully meet PLOS ONE’s publication criteria as it currently stands. Therefore, we invite you to submit a revised version of the manuscript that addresses the points raised during the review process.

Please provide necessary information in Methods and Figure Legend.

Please have careful proofreading to reduce grammatical errors. 

Please adequately address the reviewers's concerns.

A rebuttal letter that responds to each point raised by the academic editor and reviewer(s). You should upload this letter as a separate file labeled 'Response to Reviewers'.

A marked-up copy of your manuscript that highlights changes made to the original version. You should upload this as a separate file labeled 'Revised Manuscript with Track Changes'.

An unmarked version of your revised paper without tracked changes. You should upload this as a separate file labeled 'Manuscript'.

We look forward to receiving your revised manuscript.

Kind regards,

Meijing Wang, MD

Academic Editor

PLOS ONE

Journal Requirements:

Thank you for your professional advice, we have made changes to the format of the article.

Thanks to your professional opinion, we have added Anaesthesia and Euthanasia in Mice.

Thanks to your professional input, we have removed the funds from the manuscript and submitted the online funds as required.

   "Funding Title: Key Technology Research Program (Hebei Provincial Health Committee)

Research Topic: Application Study of Flow Cytometry Fluorescent Immunodetection of Cytokines in Patients with Malignant Hematologic Diseases

Funding Number: 20221410"

Thank you for your professional opinion, we have filed that statement.

5. In the online submission form, you indicated that "All data generated or analyzed during this study are included in this published article. Additional datasets analyzed during the current study are available from the corresponding author upon reasonable request."

Thank you for your professional opinion, all data generated or analysed in this study are provided as Supplementary Documents and we have provided the data as Table S1.

Thank you for your professional input, we have provided ORCID as requested.

Thank you for your professional opinion, we have provided the ethical documents as well as the ethical statement as required.

Thank you for your professional opinion, we have made additional notes.

9. PLOS ONE now requires that authors provide the original uncropped and unadjusted images underlying all blot or gel results reported in a submission’s figures or Supporting Information files. This policy and the journal’s other requirements for blot/gel reporting and figure preparation are described in detail at https://journals.plos.org/plosone/s/figures#loc-blot-and-gel-reporting-requirements and https://journals.plos.org/plosone/s/figures#loc-preparing-figures-from-image-files. When you submit your revised manuscript, please ensure that your figures adhere fully to these guidelines and provide the original underlying images for all blot or gel data reported in your submission. See the following link for instructions on providing the original image data: https://journals.plos.org/plosone/s/figures#loc-original-images-for-blots-and-gels.   

Thank you for your professional opinion, we have submitted the original strip images for the study as well as the raw data.

Reviewers' comments:

Reviewer's Responses to Questions

Comments to the Author

1. Is the manuscript technically sound, and do the data support the conclusions?

Reviewer #1: Partly

Reviewer #2: Partly

2. Has the statistical analysis been performed appropriately and rigorously? 

Reviewer #1: No

Reviewer #2: Yes

3. Have the authors made all data underlying the findings in their manuscript fully available?

Reviewer #1: Yes

Reviewer #2: Yes

4. Is the manuscript presented in an intelligible fashion and written in standard English?

Reviewer #1: No

Reviewer #2: Yes

5. Review Comments to the Author

Reviewer #1: In this manuscript, the authors investigated the role of the NOTCH1 signaling pathway in DLBCL-related heart dysfunction. They found that NOTCH1 inhibition decreased DLBCL cell proliferation, reduced the inflammatory response, and attenuated myocardial fibrotic remodeling and infarction severity following ischemic injury. However, there are several issues that need to be addressed:

1. The manuscript contains numerous writing errors and requires significant improvement in scientific writing.

Thank you for your professional input, we have asked an English professional to touch up and revise the article.

2. Detailed methods regarding the heart ischemic injury model need to be included.

Thanks to your professional opinion, we have described in detail the detailed methodology of the cardiac ischaemic injury model.

3. The immunofluorescence staining of tissue does not appear to be of heart tissue in Figure 3c.

Thank you for your professional input, we have revised and added to this section.

4. Several important details are missing, such as the dose of the STAT3 inhibitor and the sample number. Additionally, the efficiency of shRNA is not shown.

Overall, this manuscript is not well written and lacks necessary information. The conclusions are not well supported by the provided data. Therefore, I recommend a major revision to address these issues.

Thank you for your professional opinion, we have added RT-qPCR experiments for validation and added other experiments to validate the conclusions of the article.

Reviewer #2: In this study, Zhang et al. investigated the role of the NOTCH1 signaling pathway in the pathogenesis of DLBCL-related heart disease. Through data mining and bioinformatic analysis of DLBCL and myocardial infarction GEO databases, they identified the activation of signaling pathways including NOTCH, STAT3, and EZH2, underscoring their pivotal roles in the pathogenesis of DLBCL-related heart disease. However, several issues and concerns need to be addressed:

1. The authors should justify the rationale for comparing the DLBCL and myocardial infarction GEO databases to identify signaling pathways contributing to the pathogenesis of DLBCL-related heart disease.

Thank you for your professional opinion, we investigated the signalling pathways of pathogenesis by taking the intersection of the differential genes of the DLBCL dataset GSE23501 and the myocardial infarction dataset GSE48060, and enriching the intersecting genes for pathway analysis.

2. For the bioinformatic analysis, the authors need to describe the criteria for selecting the database and how the samples were categorized into reference and observation groups.

Thank you for your professional opinion, as GEO database is chosen because it covers data from many fields such as oncology and non-oncology and has high data quality and rich data types. The datasets were grouped according to the STATUS in the dataset. The myocardial infarction dataset GSE48060 includes 31 myocardial infarction patient samples and 21 control group samples; the DLBCL dataset GSE23501 includes 69 DLBCL-related samples, which are grouped according to the STATUS in the dataset, and the samples are classified into 7 reference groups and 62 observation groups.

3. The co-expression scatter plots in Figure 1 G-J do not support the claimed correlations between RBPJ and EZH2, EZH2 and H3K27, and STAT3 and JMD3, except NOTCH1 and RBPJ.

Thanks to your professional opinion, we have modified the correlation analysis for NOTCH1 and STAT3, NOTCH1 and EZH2, the correlation coefficient of NOTCH1 and STAT3 is 0.646, and the correlation coefficient of NOTCH1 and EZH2 is -0.543.

4. In Figure 5 A-D, the results for Static treatment alone should be presented to conclusively determine the synergistic effects of NOTCH and STAT3 signaling inhibition on fibroblast growth.

Thanks to your professional advice, we have added experiments in this section using STAT3 agonists and inhibitors to treat the study cells individually and set up a control group to determine the synergistic effect of NOTCH and STAT3 signalling inhibition on fibroblast growth.

5. The manuscript appears to be prepared hastily, containing numerous grammatical errors. Additionally, the figure legends lack detail and do not always correspond to the figures presented. For instance, in the legend for Figure 1, there is no distinction between panels 1A-a and 1A-b; there is no description for Figures 1D and 1E. Furthermore, Figure 1C contains two panels both labeled as Figure 1C-a. Also, many abbreviations are used without being defined upon first use.

Thank you for your professional input, we have revised and detailed the pictures and legends.

6. PLOS authors have the option to publish the peer review history of their article (what does this mean?). If published, this will include your full peer review and any attached files.

Thank you for your professional opinion, we choose no.

Do you want your identity to be public for this peer review? For information about this choice, including consent withdrawal, please see our Privacy Policy.

Reviewer #1: No

Reviewer #2: No

[NOTE: If reviewer comments were submitted as an attachment file, they will be attached to this email and accessible via the submissi

---

## [Decision Letter · Decision Letter 1]

1 Oct 2024

PONE-D-24-21738R1NOTCH1 promotes the elevation of GM-CSF and IL-6 through the EZH2/STAT3 pathway to facilitate the fibrotic state of the myocardium in DLBCLPLOS ONE

Dear Dr. Zhang,

Thank you for submitting your manuscript to PLOS ONE. After careful consideration, we feel that it has merit but does not fully meet PLOS ONE’s publication criteria as it currently stands. Therefore, we invite you to submit a revised version of the manuscript that addresses the points raised during the review process.

The manuscript has been significantly improved. However, please add sample size in figure legends and adequately address the Review #1's concerns.

We look forward to receiving your revised manuscript.

Kind regards,

Meijing Wang, MD

Academic Editor

PLOS ONE

Journal Requirements:

Reviewers' comments:

Reviewer's Responses to Questions

**Comments to the Author**

1. If the authors have adequately addressed your comments raised in a previous round of review and you feel that this manuscript is now acceptable for publication, you may indicate that here to bypass the “Comments to the Author” section, enter your conflict of interest statement in the “Confidential to Editor” section, and submit your "Accept" recommendation.

Reviewer #1: (No Response)

Reviewer #2: All comments have been addressed

2. Is the manuscript technically sound, and do the data support the conclusions?

Reviewer #1: Yes

Reviewer #2: Yes

3. Has the statistical analysis been performed appropriately and rigorously? 

Reviewer #1: Yes

Reviewer #2: Yes

4. Have the authors made all data underlying the findings in their manuscript fully available?

Reviewer #1: Yes

Reviewer #2: Yes

5. Is the manuscript presented in an intelligible fashion and written in standard English?

Reviewer #1: Yes

Reviewer #2: Yes

6. Review Comments to the Author

Reviewer #1: The updated results, such as experiments with STAT3 agonists and inhibitors, as well as the efficiency of shRNA, are missing from the revised version. Additionally, the sample size should be included.

Reviewer #2: (No Response)

7. PLOS authors have the option to publish the peer review history of their article (what does this mean?). If published, this will include your full peer review and any attached files.

Reviewer #1: No

Reviewer #2: **Yes: **Wenjun Zhang

---

## [Author Response · Author response to Decision Letter 1]

9 Dec 2024

Thank you for your professional opinion that all comments have been addressed.Thank you for your professional opinion that experiments have to be rigorously conducted and the data to support those conclusions.Thanks to your professional advice, the statistical analysis was properly and rigorously executed.Thank you for your professional opinion, we completely provided all potential discovery data.Thanks to your professional opinion, the manuscript is presented in an understandable way and written in standard English.Thanks to your expert opinion, we have added experiments with STAT3 agonists and inhibitors in Figure 5. And added the efficiency of shRNA in the results of Figure 2. In addition the sample size per group we have added to the figure notes.Thank you for your professional opinion, we opted for public peer review.

---

## [Decision Letter · Decision Letter 2]

18 Dec 2024

NOTCH1 promotes the elevation of GM-CSF and IL-6 through the EZH2/STAT3 pathway to facilitate the fibrotic state of the myocardium in DLBCL

PONE-D-24-21738R2

Dear Dr. Zhang,

We’re pleased to inform you that your manuscript has been judged scientifically suitable for publication and will be formally accepted for publication once it meets all outstanding technical requirements.

Kind regards,

Meijing Wang, MD

Academic Editor

PLOS ONE

Additional Editor Comments (optional):

Reviewers' comments:

Reviewer's Responses to Questions

**Comments to the Author**

1. If the authors have adequately addressed your comments raised in a previous round of review and you feel that this manuscript is now acceptable for publication, you may indicate that here to bypass the “Comments to the Author” section, enter your conflict of interest statement in the “Confidential to Editor” section, and submit your "Accept" recommendation.

Reviewer #1: All comments have been addressed

2. Is the manuscript technically sound, and do the data support the conclusions?

Reviewer #1: Yes

3. Has the statistical analysis been performed appropriately and rigorously? 

Reviewer #1: Yes

4. Have the authors made all data underlying the findings in their manuscript fully available?

Reviewer #1: Yes

5. Is the manuscript presented in an intelligible fashion and written in standard English?

Reviewer #1: Yes

6. Review Comments to the Author

Reviewer #1: (No Response)

7. PLOS authors have the option to publish the peer review history of their article (what does this mean?). If published, this will include your full peer review and any attached files.

Reviewer #1: **Yes: **Xiaohui Wang

---

## [Editor Report · Acceptance letter]

8 Jan 2025

PONE-D-24-21738R2 

PLOS ONE

Dear Dr. Zhang, 

I'm pleased to inform you that your manuscript has been deemed suitable for publication in PLOS ONE. Congratulations! Your manuscript is now being handed over to our production team.

Kind regards, 

on behalf of

Dr. Meijing Wang 

Academic Editor

PLOS ONE